atmospheric science/climatology

carbon dioxide, global warming, experiment, $CO_2$, atmospheric temperature

**Author for correspondence:**
Yiannis A. Levendis
e-mail: y.levendis@neu.edu

# A simple experiment on global warming

Yiannis A. Levendis, Gregory Kowalski, Yang Lu
and Gregory Baldassarre

Mechanical and Industrial Engineering Department, Northeastern University, Boston, MA 02115, USA

 YAL, 0000-0002-8158-2123

A simple experiment has been developed to demonstrate the global warming potential of carbon dioxide ($CO_2$) gas in the Earth's atmosphere. A miniature electric resistance heating element was placed inside an inflatable balloon. The balloon was filled with either air or $CO_2$. Whereas the $CO_2$ partial pressure on the earth's atmosphere is approximately $4 \times 10^{-4}$ atm, in this experiment, a high partial pressure of $CO_2$ (1 atm) was used to compensate for the short radiation absorption path in the balloon. The element was heated to approximately 50°C, the power was then switched off and the element's cooling trends in air and in $CO_2$ were monitored. It took a longer time to cool the heating element back to ambient temperature in $CO_2$ than in air. It also took longer times to cool the element in larger size balloons and in pressurized balloons when they were filled with $CO_2$. To the contrary, the balloon size or pressure made no difference when the balloons were filled with air. A simple mathematical model was developed, and it confirmed that the radiative heat loss from the element decreased significantly in $CO_2$. This investigation showed that the cooling rate of an object, with surface temperature akin to temperatures found on Earth, is reduced in a $CO_2$-rich atmosphere because of the concomitant lower heat loss to its environment.

## 1. Introduction

A benchtop experiment and analysis that qualitatively illustrates climate change mechanisms using changes in the cooling rate from an Earth-like object immersed in a radiative participating medium is presented. Global warming and climate change are terms used for the observed rise in the average temperature of the Earth's climate system and its related effects on the environment [1,2]. Since the advent of the Industrial Revolution, in the late nineteenth century, the concentration of carbon dioxide ($CO_2$) in the atmosphere has been rising, first mildly and then sharply (by 30%) over the last 50 years [3]. This rise has been related to anthropogenic generation from burning fossil fuels and deforestation. At the same time, multiple lines of

scientific evidence show that the Earth's climate system is warming [4–6]. It is also known that $CO_2$ absorbs far-infrared radiation (at wavelengths of 12–20 μm) that is emitted from moderate-temperature bodies, like the Earth's surface. The Earth's average surface temperature in recent times is also increasing [6]. $CO_2$ has been assigned a global warming potential (GWP) of unity. There are other substances that have much higher GWPs, such as the man-made choro-fluoro-refrigerants, nitric oxide and methane; however, those are found in the atmosphere at much lower concentrations than $CO_2$. Water vapour absorbs terrestrial radiation like the other greenhouse gases and produces a warming effect. However, there is also a cooling effect. As the water vapour rises in the atmosphere, it condenses and forms clouds. Clouds reflect solar radiation and reduce the heating of the earth. It has been reported that water vapour is the dominant contributor to the global greenhouse effect (approx. 50% of the effect), followed by clouds (approx. 25%) and then $CO_2$ with approx. 20%. Other absorbers play minor roles [7]. Nevertheless, as the water cycle of the atmosphere is a naturally occurring phenomenon, this work is not focusing on its effects on global warming.

A great deal of research has been conducted in the past on the effects of $CO_2$ (and other greenhouse gases, such as methane, nitric oxide, ozone and man-made refrigerants) on the warming of the earth's surface and atmosphere, from John Tyndall's seminal experiments to recent atmospheric modelling [7–21]. Most of the research, however, has involved complicated theories and mathematical models that are beyond the grasp of ordinary people with basic scientific skills who are curious to explore this issue and make informed decisions as concerned citizens of different countries. There are only a few laboratory-scale experimental demonstrations that illustrate separate components of the heat transfer in the atmosphere [22,23]. To fill this gap, a basic benchtop experiment was designed to include the combined heat transfer mechanisms that affect the cooling rate of the Earth, and it is complemented with a theoretical analysis. Both studies were conducted to illustrate the physics of heat transfer from a small electric heater, of a temperature akin to that of Earth, when it is exposed to atmospheres composed of high concentration of $CO_2$. The effect this major greenhouse gas has on the heat transfer rate from this heater is observed through its cooling rate. The complicating factors of water vapour, clouds, aerosols, water bodies, etc., have not been taken into account. While the scale of the benchtop experiment precludes an exact analogy with global warming, because of the combined radiative and convective heat transfer mechanism, the experiments do demonstrate the effects of the presence of $CO_2$ on outgoing radiation from the Earth's surface. Moreover, the theoretical analysis confirms the cooling behaviour and the expected magnitude of different cooling rates with and without $CO_2$.

In this experiment, Earth is approximated with the small heater enclosed in a much larger gas-filled spherical balloon, which approximates the Earth's atmosphere. The balloon is filled with either air or $CO_2$. The heater is deliberately brought to Earth-like temperatures. The heater's surrounding gas also heats up and a negative temperature gradient develops in the gas inside the balloon, in the outward radial direction. This temperature gradient in the gas is akin to the negative temperature gradient with increasing altitude inside Earth's troposphere. Upon reaching a predetermined temperature (50°C or 323 K), the electricity is turned off and the heater is allowed to cool by transferring energy (heat) to the balloon environment. Heat transfer takes place by convection to the gas inside the balloon and by radiation to its surroundings in the far-infrared wavelength spectrum. This is similar to Earth's case. A portion of the long-wavelength radiation is absorbed by the gas inside the balloon when it contains infrared radiation (IR) absorbing molecules, such as $CO_2$. Another portion of the long-wave radiation is transmitted through the gas and then it is mostly transmitted outwards through the membrane of the balloon. A fraction is probably absorbed by the balloon. This is analogous to what occurs in the Earth's atmosphere, where radiation emitted by the surface of the Earth is absorbed by IR-absorbing molecules in the atmosphere, clouds or other aerosols [24]. A fraction of the emitted radiation from the Earth's surface escapes to outer space through 'the atmospheric window'.

The Earth's atmosphere emits radiation into a spherical shell. A portion of this emitted radiation is directed back to Earth and a portion is directed to outer space. A similar heat transfer process occurs in the heater/balloon experiment. Although this gas emission differs from the radiation emitted by a black body, it is often found convenient in radiation calculations to treat gas emissions as a black body at some lower effective temperature. This temperature is called the effective sky temperature [25], and it is the temperature of the IR-absorbing species in the atmosphere. The radiation emitted from these species to space will be lower than that emitted from the Earth's surface to space, had those IR species been absent. This is because the temperature lapse rate (temperature gradient) in the atmosphere is negative ($-6.5$ K km$^{-1}$); hence, the IR-absorbing species are at a lower temperature than the Earth's surface. This produces a planetary energy imbalance, which leads to warming [26].

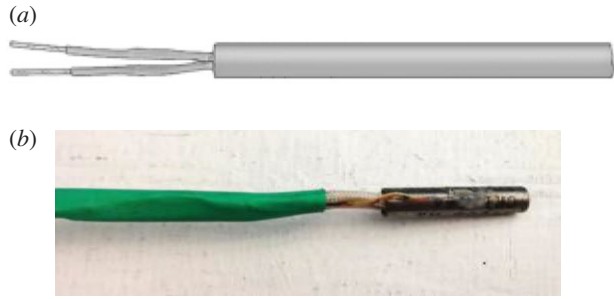

**Figure 1.** (*a*) Cartridge heater. (*b*) Thermocouple epoxied to the heating element.

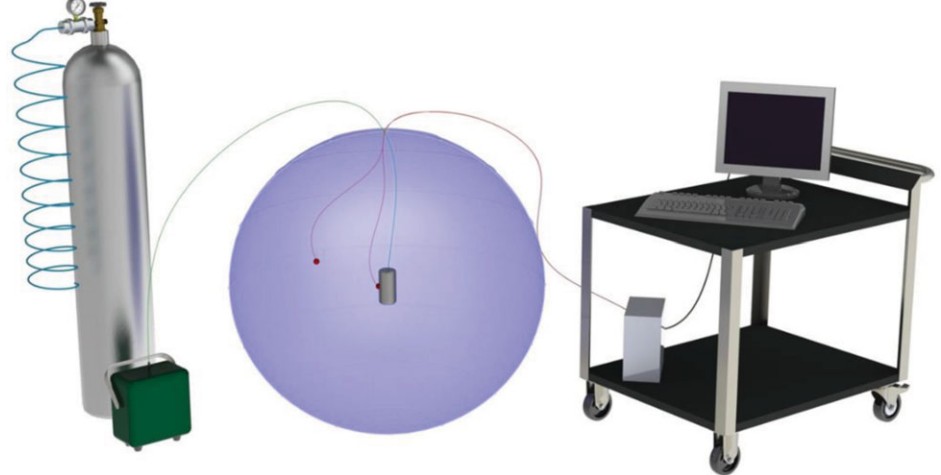

**Figure 2.** A schematic of the experimental set-up consisting of an inflatable balloon, a miniature electric resistance heating element, temperature sensors and instrumentation to record real-time temperature [27].

## 2. Experimental design

A miniature cartridge, 120 V heating element (figure 1*a*) with a diameter of 0.635 cm and a length of 2.54 cm, was used in these experiments to approximate Earth. Both the Earth and this object lose heat by emitting long-wavelength radiation and by convection. The power output of the heating element was controlled by a rheostat and its temperature was monitored by an attached thermocouple (figure 1*b*).

The atmosphere of Earth, containing $CO_2$, was approximated by the gas contained in the balloon. The experimental set-up is illustrated in the schematic of figure 2 and the hardware is shown in figure 3.

Polymeric (PVC) membrane balloons were chosen to contain gases relevant to the Earth's atmosphere. The balloons were commercially available exercise balls with diameters of 65 cm, 75 cm and 85 cm. The heating element was inserted through a small hole punctured on the balloon and then sealed with silicone epoxy. Temperature measurements were conducted with a nickel-alloy T-type thermocouple (standard: ±1.0 K or ±0.75%; special limits of error: ± 0.5 K or 0.4%). $CO_2$ or air were introduced into the balloon from laboratory gas cylinders using Matheson regulators. The pressure in the balloon was measured with a U-tube manometer. The temperature sensors were monitored by a National Instruments Sensor DAQ Data Acquisition unit, connected to a USB port of a desktop computer. Data were also collected through a traditional terminal box and a data acquisition card but this combination was more susceptible to electronic noise. The LABVIEW software was used for real-time data taking. All experiments were replicated in triplicate by different users using different balloons. In each experiment, the heating element was brought expediently to a higher temperature (323 K) than the ambient temperature and it was then allowed to cool. Experimental data are available in [28].

## 3. Experimental results

Experimental results, plotted in figure 4, showed that upon switching the electric heating element off, its cooling rate was slower in $CO_2$ than in air. For example, in the smallest balloon tested (65 cm), the two

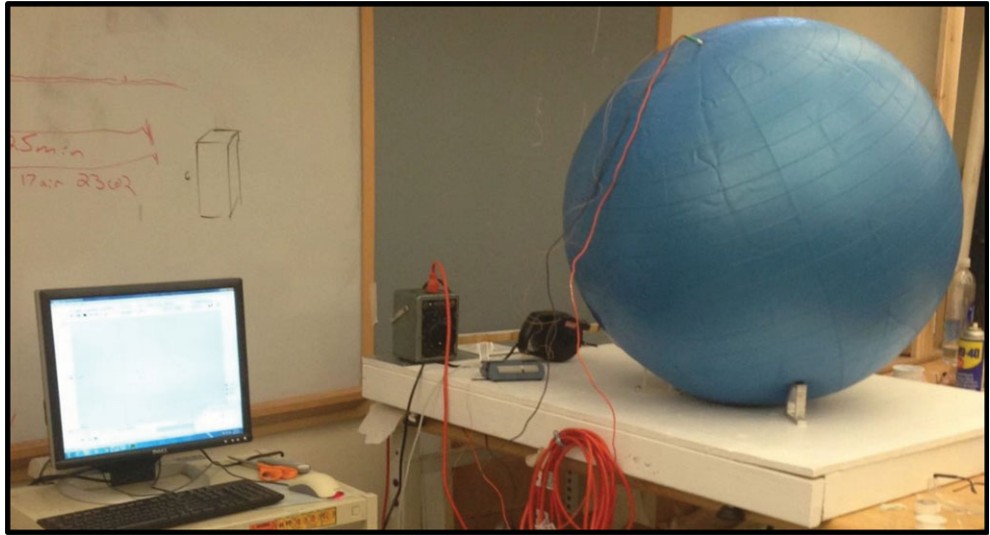

**Figure 3.** A photograph of the experimental hardware consisting of a balloon, a miniature electric resistance heating element, temperature sensors and instrumentation to record temperature [22].

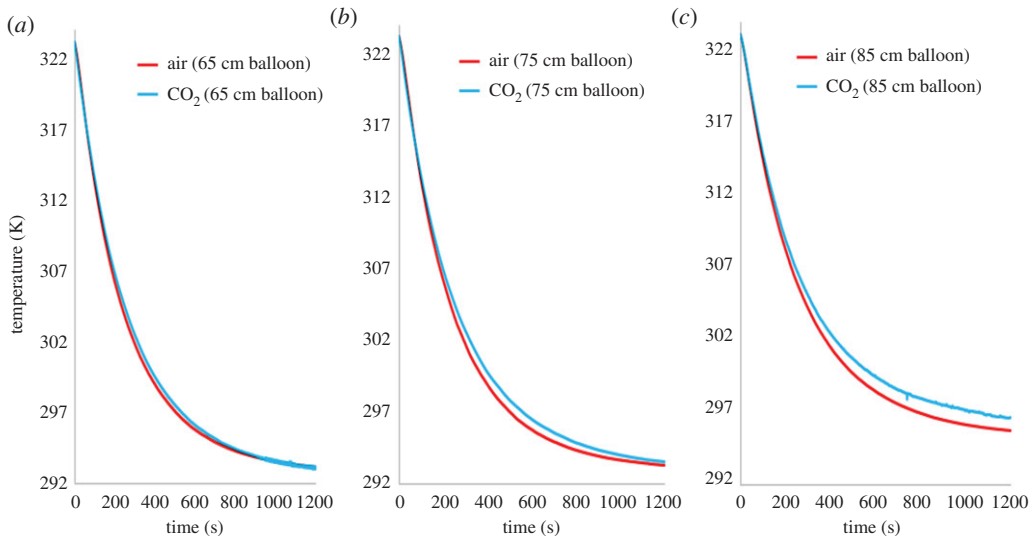

**Figure 4.** Experimentally obtained cooling curves of the electric resistance heating element (initially heated to 323 K) in three different diameter balloons ((*a*) 65 cm; (*b*) 75 cm; (*c*) 85 cm).

cooling curves were separated by a small temperature difference of 0.5 K at 6.5 min, which is at the thermocouple special limit of error. However, in the 75 cm balloon, the temperature difference at 6.5 min increased to a more significant 1.5 K, whereas in the 85 cm balloon, the corresponding temperature difference increased further to 2 K; figure 4. These results demonstrate that larger amounts of $CO_2$, in this case, thicker layers of the gas, decrease the cooling rate of the heating element. This is expected as the radiative cooling of an object is a function of both the absorption path length and the concentration of the absorbing surrounding medium [29,30].

To account for any day-to-day variations in the ambient temperature of the laboratory, normalized temperature differences were also calculated using equation (3.1), and are plotted in figure 5:

$$\Delta T = \frac{T(\text{time}) - T_{\text{ambient}}}{T(\text{initial}) - T_{\text{ambient}}}. \tag{3.1}$$

These results illustrate the temperature difference between the heating element and the ambient relative to the maximum temperature possible in the experiment. A value of 1 refers to the maximum temperature difference, i.e. the initial point, and a value of 0 refers to the final state, i.e. the point of the element's thermal equilibrium with the ambient temperature. Any contributions from variations in

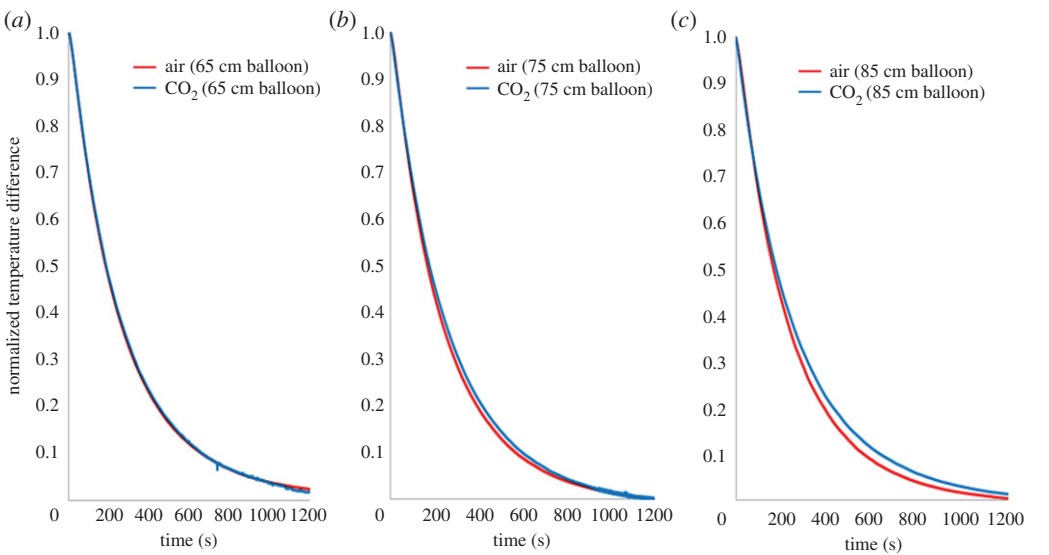

**Figure 5.** Experimentally obtained temperature difference cooling curves of the electric resistance heating element (initially heated to 323 K) in three different diameter balloons ((a) 65 cm; (b) 75 cm; (c) 85 cm).

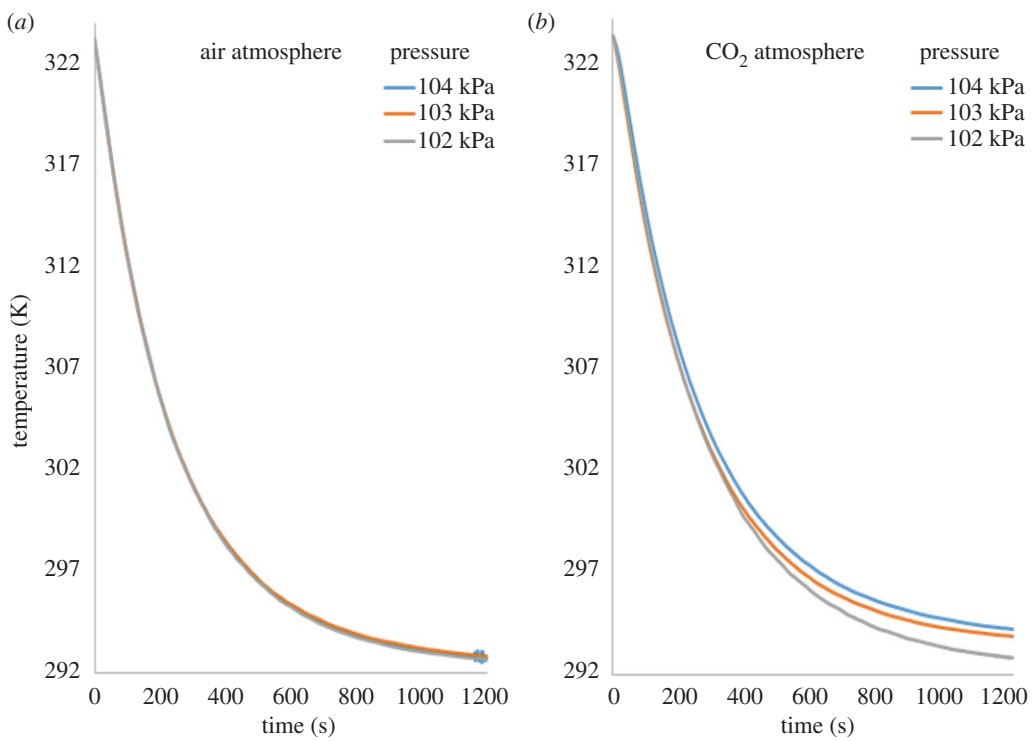

**Figure 6.** Experimentally obtained cooling curves of the electric resistance heating element (initially heated to 323 K) in the largest of the balloons (85 cm) pressurized to three different pressures: 102, 103 and 104 kPa; (a) in air, (b) in $CO_2$.

the room temperature are eliminated in these results. The observed differences between the cooling curves for the $CO_2$ gas, as compared to the air within the balloon, are clearly evident in figure 5. The cooling of the heating element is noticeably impeded in $CO_2$, and this impedance increased with increasing the quantity, in this case, the thickness of the layer, of this gas.

Besides increasing the size of the balloon, the gas was pressurized to further increase the mass of $CO_2$ therein, albeit only mildly in the case of these rubber membrane balloons. The effect of the gas pressure in the balloon on the cooling rate of the heater was assessed in the second series of experiments. The gas pressure was measured with a manometer, connected to a two-way valve at the inlet of the balloon. To preserve the integrity of the balloon, pressurization was only conducted in the narrow pressure range of 102–104 kPa. The results are shown in figure 6.

**Table 1.** Constants of electric cartridge heater (SS 304).

| | |
|---|---|
| surface area, $A$ ($m^2$) | $5.07 \times 10^{-4}$ |
| volume, $V$ ($m^3$) | $8.04 \times 10^{-7}$ |
| characteristic length = equivalent sphere diameter = $2r_e$ (m) | $1.58 \times 10^{-3}$ |
| emissivity, $\varepsilon$ | 0.97 |
| thermal conductivity, $k$ (W $m^{-1}$ $K^{-1}$) | 16.2 |
| specific heat, $c$ (J $kg^{-1}$ $K^{-1}$) | 480 |
| density, $\rho$ (kg $m^{-3}$) | 8055 |

It can be seen that whereas in air there was no discernible effect of pressure on the cooling curve of the heating element, in $CO_2$, the differentiation of the cooling curves is evident. Increasing the pressure in the balloon, i.e. augmenting the mass of $CO_2$ and thus its optical thickness, increased the absorption of the radiated energy by the gas from the cooling element. As a result, its cooling was clearly impeded.

# 4. Theoretical calculations

The experimental study was complemented with a theoretical analysis of the expected cooling process of the heating element. In this analysis, the lumped capacitance method outlined in [29] was used because the spatial temperature gradients within the heater were considered to be negligible. For simplicity, the short cylindrical heater was treated as a sphere with an equivalent radius, $r_e$, defined by equation (4.1):

$$r_e = 3\left(\frac{V_{\text{heater}}}{A_{\text{heater}}}\right), \tag{4.1}$$

where $V_{\text{heater}}$ and $A_{\text{heater}}$ are the volume and the surface area of the heater, respectively. Cooling of the heater is by convective and radiative heat losses. This calculation aims to assess the effects of $CO_2$, in contrast with air, on both the convective and radiative energy losses from the heating element of mass $m_{\text{heater}}$ and heat capacity $c_{\text{heater}}$. The heater is used here to simulate the Earth. The cooling process is described using a transient energy balance on the heater system, equation (4.2):

$$(m_{\text{heater}}c_{\text{heater}})\frac{T_{\text{heater}}}{dt} = \rho V c_{\text{heater}}\frac{dT_{\text{heater}}}{dt} = -\dot{Q}_{\text{convection}} - \dot{Q}_{\text{radiation}}. \tag{4.2}$$

The parameters for the heater are summarized in table 1.

## 4.1. Convective energy losses

The rate of heat transfer via natural convection is given by the equation:

$$\dot{Q}_{\text{convection}} = h_{\text{conv}}A_{\text{heater}}(T_{\text{heater}} - T_{\text{gas}}), \tag{4.3}$$

where $\dot{Q}_{\text{convection}}$ is the amount of heat transfer by convection, $h_{\text{conv}}$ is the convective heat transfer coefficient, $T_{\text{gas}}$ is the surrounding gas temperature in the balloon and $T_{\text{heater}}$ is the temperature of the heater. The convective heat transfer coefficient is calculated using the following Nusselt–Rayleigh number correlation:

$$\text{Nu} = 2 + \frac{0.589\,\text{Ra}^{1/4}}{[1 + (0.469/\text{Pr})^{9/16}]^{4/9}} = \frac{h_{\text{conv}}(2r_e)}{k_{\text{gas}}}, \tag{4.4}$$

where the Rayleigh number, Ra, is defined as:

$$\text{Ra} = \text{Pr}\,\text{Gr} = \frac{g\,\beta}{\nu\,\alpha_d}(T_{\text{heater}} - T_{\text{gas}})\,(2r_e)^3. \tag{4.5}$$

The parameters Pr and Gr are the Prandtl and Grashof numbers, respectively; $k_{\text{gas}}$ is the thermal conductivity of the fluid, $\nu$ is the kinematic viscosity of the gas, and $\alpha_d$ is the thermal diffusivity of the

**Table 2.** Constants for calculating convective heat losses in air and carbon dioxide.

| | air | $CO_2$ |
|---|---|---|
| gravitational acceleration, $g$ (m s$^{-2}$) | 9.816 | |
| surface temperature of the heater, $T_{heater}$ (K) | 323.15 | |
| initial temperature of gas $T_{gas}$ (K) | 296.15 | |
| final temperature of gas $T_{gas\ final}$ (K) | 297.15 | |
| thermal expansion coefficient, $\beta$ | $3.388 \times 10^{-3}$ | |
| thermal conductivity of the gas, $k_{gas}$ (W m$^{-1}$ K$^{-1}$) | 0.0295 | 0.0166 |
| Prandtl number, Pr | 0.700 | 0.766 |
| kinematic viscosity, $v$ (m$^2$ s$^{-1}$) | $15.9 \times 10^{-6}$ | $8.4 \times 10^{-6}$ |
| thermal diffusivity, $\alpha_d$ (m$^2$ s$^{-1}$) | $22.5 \times 10^{-6}$ | $11.0 \times 10^{-6}$ |
| Nusselt number, Nu | 2.72 | 3.13 |
| convective heat transfer coefficient, $h_{conv}$ (W m$^{-2}$ K$^{-1}$) | 45.6 | 32.8 |

gas. The convective heat transfer parameters are summarized in table 2. The temperatures summarized in table 2 are used in the calculations and are representative of the experimental values.

## 4.2. Radiative energy losses

The gas is treated as a participating medium to calculate the radiation loss from the heater to the surrounding gas. The analysis is based on the gas emissivity and absorptivity, $\varepsilon_{gas}$ and $\alpha_{gas}$, respectively, as reported by [26–31] and is summarized in equation (4.6):

$$\dot{Q}_{radiation} = \sigma \varepsilon_{heater} A_{heater} (1 - \alpha_{gas})(T_{heater}^4 - T_{gas}^4)$$
$$= h_{rad} A_{heater} (T_{heater} - T_{gas}), \tag{4.6}$$

where,

$$h_{rad} = \sigma \varepsilon_{heater} (1 - \alpha_{gas})(T_{heater} + T_{gas})(T_{heater}^2 + T_{gas}^2). \tag{4.7}$$

The emissivity of the heater was assumed to be 0.85 for a lightly oxidized metallic surface. The gas emissivity and gas absorptivity are determined from Hottel's charts [24–26] in terms of the product of the gas pressure times the equivalent gas thickness, $PL_e$, and a pressure correction factor, $C$:

$$\alpha_{gas} = C \left( \frac{T_{gas}}{T_{heater}} \right)^{0.65} \varepsilon_{gas}. \tag{4.8}$$

The literature contains simplified approaches for describing the effect of participating media on the radiative transfer from surfaces [29–31]. These methods use Hottel's method as a simplified approach to determine the radiant heat flux between an absorbing gas and an adjoining surface. A hemispherical gas mass was assumed at temperature $T_{gas}$ and a surface element $dA$, located at the centre of the hemisphere's base. Emission of the gas per unit area of the base of the surface was expressed as $E_{gas} = \varepsilon_{gas} \sigma T_{gas}$. The emissivity $\varepsilon_{gas}$ was correlated in terms of both pressure and temperature of the gas, the partial pressure of the radiating species and the radius of the hemisphere. As a first-order approximation, the gas was assumed to be at a uniform temperature. This is not quite the case for the experiment, which has a small negative temperature gradient. This is also the case for the atmosphere. Hottel extended the results to other gas geometries using the concept of mean beam length, $L_{m-b}$, interpreted as the radius of a hemispherical gas mass whose emissivity is equivalent to that of geometry of interest [19–31]. The equivalent length, $L_{m-b}$, for a spherical segment is related to its spherical diameter, [29–31]:

$$L_{m-b} = 3.66 \frac{V_{chamber}}{A_{chamber}} = 3.66(0.167)d = 0.61\, d. \tag{4.9}$$

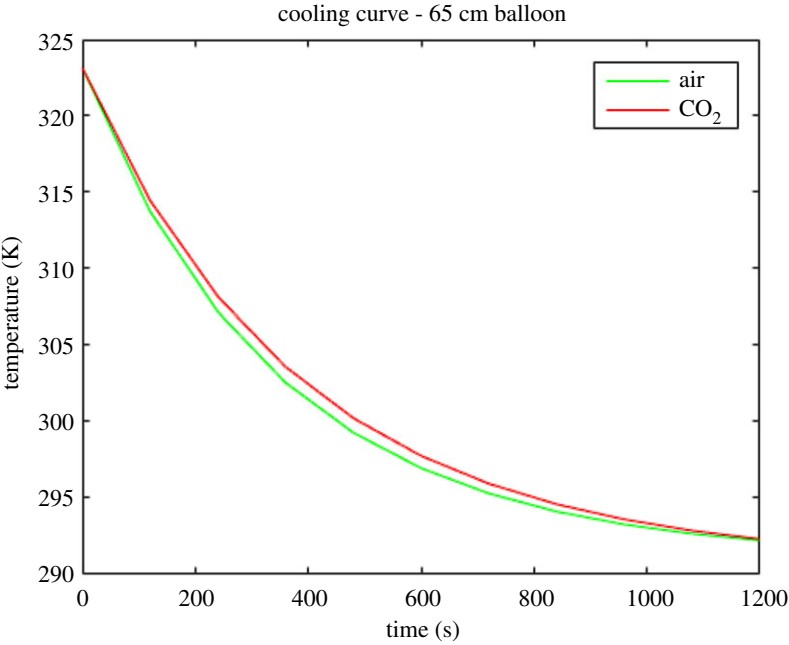

**Figure 7.** Theoretically obtained cooling curves of the electric resistance heating element (initially heated to 323 K) in the 65 cm balloon.

**Table 3.** Constants used for calculating absorptivity and emissivity of carbon dioxide.

| balloon diameter $d$ (cm) | 65 cm | 75 cm | 85 cm |
| --- | --- | --- | --- |
| mean beam length, $L_{m-b}$ (m) | 0.396 | 0.457 | 0.518 |
| $PL_e$ (ft-atm) | 1.3 | 1.5 | 1.7 |
| $C(P)$ | 1 | 1 | 1 |
| emissivity of $CO_2$, $\varepsilon_{gas}$ | 0.15 | 0.16 | 0.17 |
| absorptivity of $CO_2$, $\alpha_{gas}$ | 0.146 | 0.156 | 0.166 |
| radiative heat transfer coefficient, $h_{rad}$ (W m$^{-2}$ K$^{-1}$) | 4.90 | 4.84 | 4.78 |

Air is considered as a transparent gas; thus, its emissivity $\varepsilon_{gas} = 0$. The emissivity for $CO_2$ gas is found using the initial experimental temperature of the heating element of 323 K and the temperature of the ambient air at 296 K. Table 3 shows the constants used to determine the emissivity using Hottel's method [29,30].

The radiative constants for the $CO_2$-filled balloon are listed in table 3. The radiative heat transfer coefficient of the $CO_2$-filled balloon is lower than that of the air-filled balloon, which has a value of $h_{rad} = 5.74$ W m$^{-2}$ K$^{-1}$. The radiative heat transfer decreases, as expected, as the mean path length increases with the larger $CO_2$-filled balloon diameter.

The transient energy balance on the heater system (simulating the Earth) relates the time rate of change of the heater energy stored to the radiative and convective heat transfer rates (equation (4.10)). Conduction heat transfer along the connecting wires is negligible because the lead wire cross-sectional area is small compared to the total heater area.

Substituting equations (4.3) and (4.6) for the convective and the radiative heat transfer terms into equation (4.2), we obtain:

$$\frac{dT_{heater}(t)}{dt} = \frac{-\sigma\varepsilon_{heater}A_{heater}(1 - \alpha_{gas})}{\rho V c}T_{heater}^4(t) - \frac{h_{conv}A_{heater}}{\rho V c}T_{heater}(t)$$
$$+ \frac{\sigma\varepsilon_{heater}A_{heater}(1 - \alpha_{gas})T_{gas}^4 + h_{conv}A_{heater}T_{gas}}{\rho V c}. \tag{4.10}$$

The transient response of the heater, equation (4.10), was numerically solved with the MATLAB software.

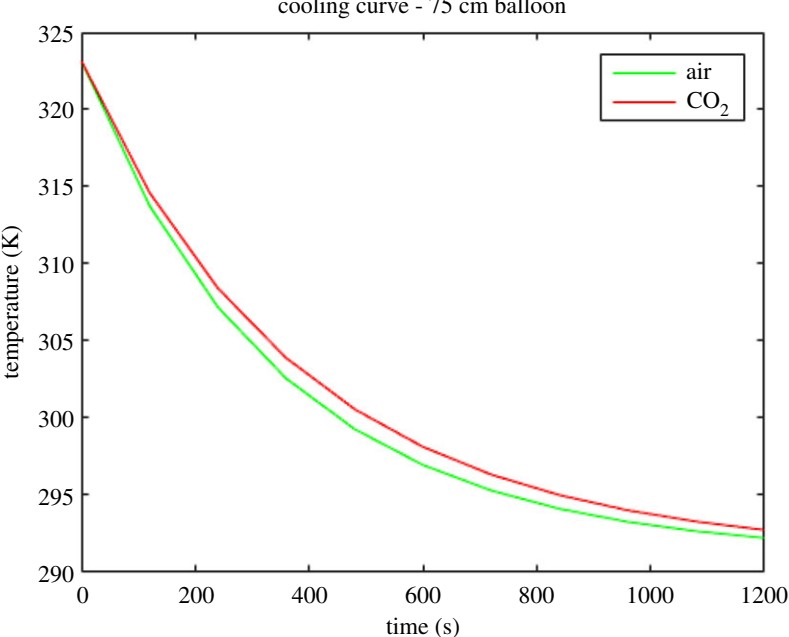

**Figure 8.** Theoretically obtained cooling curves of the electric resistance heating element (initially heated to 323 K) in the 75 cm balloon.

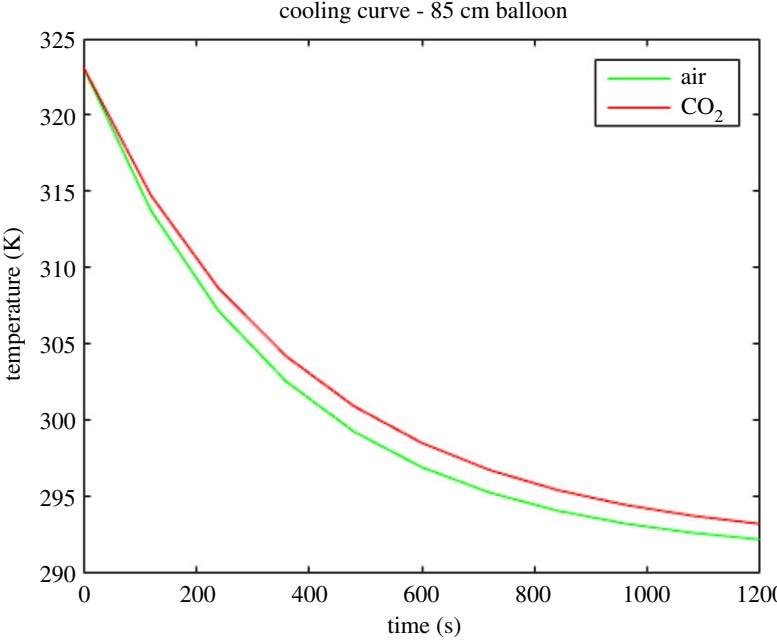

**Figure 9.** Theoretically obtained cooling curves of the electric resistance heating element (initially heated to 323 K) in the 85 cm balloon.

The Biot number (equation (4.11)) of the heating element was calculated and found to be in the order of $10^{-3}$ for both the air and $CO_2$ cases. This confirmed that the lumped parameter analysis for the combined convective and radiative heat loss is accurate:

$$B_i = \frac{(h_{conv} + h_{rad})(2r_e)}{k_{heater}}. \tag{4.11}$$

The results of the numerical solution of equation (4.10) for the 65 cm, 75 cm and 85 cm diameter balloons are shown in figures 7–9. The red lines depict the calculated cooling curves of the heating element in $CO_2$ and the green lines depict the calculated cooling curves of the heating element in the air.

The experimental and calculated results illustrate a consistent trend where the heater in the $CO_2$-filled balloon has a slower cooling rate than the heater in the air-filled balloon. The slower cooling rate is a result of a decrease in the radiative heat transfer from the simulated Earth (the heater) that is a result of the increased gas absorption in the $CO_2$-filled balloon. Moreover, it is important to note that the modelling demonstrated that the convective heat transfer from the heater is the dominant mechanism in both the air and the $CO_2$ cases. Based on the results shown in tables 2 and 3, the total energy losses in the air-filled balloon were $h_{conv} + h_{rad} = 45.6 + 5.74 = 51.3 \, W \, m^{-2} \, K^{-1}$, whereas those in the $CO_2$-filled balloon were $h_{conv} + h_{rad} = 32.8 + 4.90 = 37.7 \, W \, m^{-2} \, K^{-1}$. The convective heat loss amounted to 89% of the total energy losses of the heating element in the air-filled balloon, and 87% of the energy losses from the heating element in the $CO_2$-filled balloon. This relatively large convective loss contribution is owing to the scale limits of a benchtop experiment in not duplicating the dominance of radiation on climate change. Even so, the simulation predicts that the radiative heat loss decreased by 17% in the $CO_2$-filled balloon, as compared to that in the air-filled balloon. The large fraction of total energy loss by convection in the experiment is more consistent with the combined heat transfer mechanisms in the troposphere that contribute to the temperature profile therein. The global energy exchange between the Earth and space is dominated by the radiative heat transfer. This simple, benchtop experiment does provide a means to illustrate how the small decrease in the radiative heat transfer from the simulated Earth affects the observed cooling rate in the experiment and calculations. The combination of this experiment and analysis is a major contribution of this paper. The observed and predicted cooling rates from the heater support the hypothesis that increases in the $CO_2$ concentration in the atmosphere lead to reduced heat transfer from the Earth.

Data accessibility. Levendis, Yiannis; Kowalski, Gregory; Lu, Yang; Baldassarre, Gregory (2019), 'Global warming experiments', Mendeley data, v1: http://dx.doi.org/10.17632/hvp6c3n62z.1.

Authors' contributions. All authors have read and approved this manuscript before submission. Y.A.L. initiated the conceptual design of the experiment, oversaw the execution of the experiments, contributed to the analysis and put together the manuscript from the students' reports. G.B., while an undergraduate student, developed and constructed the initial experimental set-up as part of his senior year Capstone Engineering Design course, took and analysed data, conducted theoretical calculations and wrote a report. Y.L., while a graduate student, re-developed and constructed a new experimental set-up to verify the first set of data, collected and analysed a new set of data, improved on theoretical calculations and wrote a report. G.K. gave important input on the experimental design and execution of the experiments, contributed to the interpretation of the results and helped put together the manuscript. He also took a leading role in the theoretical analysis. All authors gave final approval for publication.

Competing interests. We have no competing interests.

Funding. No external funding supported this research. It was entirely supported by Northeastern University.

Acknowledgements. This research was supported by the Department of Mechanical and Industrial Engineering at Northeastern University. Extensive help by Amy Doyle in the experiments and in the theoretical calculations is acknowledged.

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
