## [Reviewer comments · Royal Society Open Science]

Review History

RSOS-190208.R0 (Original submission)

Review form: Reviewer 1

Is the manuscript scientifically sound in its present form?

Yes

Are the interpretations and conclusions justified by the results?

Yes

Is the language acceptable?

Yes

Is it clear how to access all supporting data?

Yes

Do you have any ethical concerns with this paper?

No

Have you any concerns about statistical analyses in this paper?

No

Recommendation?

Major revision is needed (please make suggestions in comments)

Comments to the Author(s)

These experiments appear to be accessible to an educated non-expert audience in line with the Introduction. The effect of a CO₂ environment on reducing the cooling rate of a heated immersed object below that measured in air is clear and well discussed. The analytical section is insufficiently detailed and too disjointed to achieve the same end. The Hottle method for dealing with gaseous radiation is approximate but easy to understand, however the narrative does not attempt to describe the Hottle method, even in a cursory fashion, nor to make clear how the energy balance was done and how the Hottle absorptivity, emissivity and mean beam length (or average mean beam length) fit in. My attached comments (Appendix A) point out several typographical errors as well as inconsistencies in notation.

Review form: Reviewer 2**Is the manuscript scientifically sound in its present form?**

No

Are the interpretations and conclusions justified by the results?

No

Is the language acceptable?

Yes

Is it clear how to access all supporting data?

Not Applicable

Do you have any ethical concerns with this paper?

No

Have you any concerns about statistical analyses in this paper?

No

Recommendation?

Reject

Comments to the Author(s)

This paper is a description of experiment designed to replicate the greenhouse effect and demonstrate the warming power of carbon dioxide. I look favorably upon their goal, but unfortunately their experiment does not reproduce the physics of the greenhouse effect.

The main problem with this paper is that their experiment does simulate the physics of the greenhouse effect. The greenhouse effect works because there is a temperature gradient in the atmosphere – the upper atmosphere is colder than the lower atmosphere. If you add carbon dioxide, you raise the effective radiating level of the atmosphere, so that the atmosphere is then radiating from a colder temperature. This produces a planetary energy imbalance, which leads to warming.

This is such basic stuff that I couldn't find a good peer-reviewed paper to cite. So I would point them to these two links: https://skepticalscience.com/basics_one.html or <https://www.youtube.com/watch?v=4PAbm1u1IVg>.

As a result, while their experiment gets the right answer (carbon dioxide warms the climate), it does so for the wrong reason. Thus, I don't think this paper is publishable in its present form.

A few more minor comments:

Line 54: "However, the net influence of the water vapor on atmospheric temperature is unclear."

No, water vapor is responsible for the majority of the greenhouse effect. See this paper:

<https://agupubs.onlinelibrary.wiley.com/doi/full/10.1029/2010JD014287>

I found the nomenclature of $T(\infty)$ being the initial temperature to be confusing. It seems like that should be the long-time equilibrium temperature.

Eq. 11 jumped out of nowhere. I'm not sure how you got that from the previous equations.

I feel bad being so negative and I encourage the authors to keep working on producing a good simulation of the greenhouse effect.

Decision letter (RSOS-190208.R0)

26-Jul-2019

Dear Dr Levendis:

Manuscript ID RSOS-190208 entitled "A Simple Experiment on Global Warming" which you submitted to Royal Society Open Science, has been reviewed. The comments from reviewers are included at the bottom of this letter.

In view of the criticisms of the reviewers, the manuscript has been rejected in its current form. However, a new manuscript may be submitted which takes into consideration these comments.

Please note that resubmitting your manuscript does not guarantee eventual acceptance, and that your resubmission will be subject to peer review before a decision is made.

Your resubmitted manuscript should be submitted by 23-Jan-2020. If you are unable to submit by this date please contact the Editorial Office.

on behalf of Jon Blundy (Subject Editor)
openscience@royalsociety.org

Associate Editor Comments to Author:

Thank you for your patience while the Editors sought reviewers for your paper. Regrettably, the journal has had to approach an unusually large number of potential reviewers to secure sufficient reports for the Editors to make an informed judgement on the paper.

As you'll see, both reviewers consider the paper to have a useful goal, but each provide criticisms that preclude the possibility of publishing the current version of the paper. That being said, as the comments suggest a number of ways to improve the work, we would like you to consider the reviewers' reports and make the required changes before submitting a modified manuscript for further review.

Given the scale of the changes, a regular 'revision' would not provide you sufficient time to complete the task, so we're rejecting the current version and inviting a resubmission, which will give you a longer time horizon to achieve the changes.

Please ensure you include a point-by-point response to the reviewers and also a marked-up version of the paper that allows easy tracking of the changes you've made. Good luck!

Reviewers' Comments to Author:

Reviewer: 1

Comments to the Author(s)

These experiments appear to be accessible to an educated non-expert audience in line with the Introduction. The effect of a CO₂ environment on reducing the cooling rate of a heated immersed object below that measured in air is clear and well discussed. The analytical section is insufficiently detailed and too disjointed to achieve the same end. The Hottle method for dealing with gaseous radiation is approximate but easy to understand, however the narrative does not attempt to describe the Hottle method, even in a cursory fashion, nor to make clear how the energy balance was done and how the Hottle absorptivity, emissivity and mean beam length (or average mean beam length) fit in. My attached comments point out several typographical errors as well as inconsistencies in notation.

Reviewer: 2

Comments to the Author(s)

This paper is a description of experiment designed to replicate the greenhouse effect and demonstrate the warming power of carbon dioxide. I look favorably upon their goal, but unfortunately their experiment does not reproduce the physics of the greenhouse effect.

The main problem with this paper is that their experiment does simulate the physics of the greenhouse effect. The greenhouse effect works because there is a temperature gradient in the atmosphere – the upper atmosphere is colder than the lower atmosphere. If you add carbon dioxide, you raise the effective radiating level of the atmosphere, so that the atmosphere is then radiating from a colder temperature. This produces a planetary energy imbalance, which leads to warming.

This is such basic stuff that I couldn't find a good peer-reviewed paper to cite. So I would point them to these two links: https://skepticalscience.com/basics_one.html or <https://www.youtube.com/watch?v=4PAbm1u1IVg>.

As a result, while their experiment gets the right answer (carbon dioxide warms the climate), it does so for the wrong reason. Thus, I don't think this paper is publishable in its present form.

A few more minor comments:

Line 54: "However, the net influence of the water vapor on atmospheric temperature is unclear."
 No, water vapor is responsible for the majority of the greenhouse effect. See this paper:
<https://agupubs.onlinelibrary.wiley.com/doi/full/10.1029/2010JD014287>

I found the nomenclature of $T(\infty)$ being the initial temperature to be confusing. It seems like that should be the long-time equilibrium temperature.

Eq. 11 jumped out of nowhere. I'm not sure how you got that from the previous equations.

I feel bad being so negative and I encourage the authors to keep working on producing a good simulation of the greenhouse effect.

Author's Response to Decision Letter for (RSOS-190208.R0)

See Appendices B & C.

RSOS-192075.R0

Review form: Reviewer 2

Is the manuscript scientifically sound in its present form?

No

Are the interpretations and conclusions justified by the results?

No

Is the language acceptable?

Yes

Do you have any ethical concerns with this paper?

No

Have you any concerns about statistical analyses in this paper?

No

Recommendation?

Major revision is needed (please make suggestions in comments)

Comments to the Author(s)

The paper describes a simple experiment that demonstrates that a CO₂ atmosphere cools more slowly than an O₂/N₂ atmosphere.

This is a revision of a paper that I had previously reviewed. I criticized the previous version because it did not reproduce the actual physics of the Earth's greenhouse effect. To the authors' credit, this version is improved. In particular, they have improved the discussion of the physics of what's going on in our climate system and seem to recognize the shortcomings of their experiment.

In the end, however, I still don't think this experiment reproduces the actual physics of the greenhouse effect. If you go to line 328 of the paper, where it refers to tables 2 and 3, they show that the main difference between heat loss in air and CO₂ is in the amount convective heat transport. This is not how CO₂ warms the Earth, which is mainly by modifying radiation to space (they know this, see line 336).

So I agree that they have shown that CO₂ reduces the cooling rate in their experiment, but the applicability to the present climate problem seems to me to be limited.

One way to fix this would be to put a prominent disclaimer up front in the paper that says that the mechanism here is not the same as those that are driving climate change. I think that would remove any scientific problems I have with this paper. However, this would create another problem. If this experiment is not applicable to the Earth, then what's the point of publishing the paper? Perhaps the authors could figure out a way to finesse that ... ?

Review form: Reviewer 3

Is the manuscript scientifically sound in its present form?

Yes

Are the interpretations and conclusions justified by the results?

Yes

Is the language acceptable?

Yes

Do you have any ethical concerns with this paper?

No

Have you any concerns about statistical analyses in this paper?

No

Recommendation?

Accept with minor revision (please list in comments)

Comments to the Author(s)

This manuscript describes laboratory and numerical experiments to demonstrate the impacts of presence of CO₂ in the Earth's atmosphere on the earth-atmosphere energy exchange. The authors placed a heating element in an inflatable balloon and examine the time it takes to cool the element back to ambient temperature in the presence and absence of CO₂ (akin to examining surface temperature in the earth's atmosphere). Not surprisingly, they demonstrate that the radiative heat loss from the element decreased in the presence of CO₂. As such the experiment could be regarded as a simple demonstration of the effects of the presence of CO₂ on perturbing the earth-atmosphere radiation budget and the atmosphere's subsequent warming, but not necessarily a rigorous representation of the magnitude of the warming or the complex interactions resulting in climate change. A quick search of the internet yields links to experiments which provide similar qualitative inferences on the effects of CO₂ and its greenhouse effect, for example:

<https://www.rsc.org/Education/Teachers/Resources/jesei/co2green/home.htm>

http://www.carboeurope.org/education/CS_Materials/Bernd-BlumeExperiments.pdf

Perhaps it could be argued that experiments presented here are more rigorous and controlled than the two examples above and that the authors have also developed an energy exchange mathematical model to capture and corroborate their experimental set-up. Climate change is one of the most prominent and debated societal issues currently; significant scientific and political debate exists on the magnitude and causes for the observed atmospheric warming trends. Thus, simple representations of such complex phenomena are useful to illustrate the likely impacts of rising atmospheric CO₂ concentrations. While the experiments presented do demonstrate CO₂ induced warming and the presentation of the mathematical framework is a useful contribution, I struggled to clearly identify the unique scientific contribution of this work – perhaps the authors can state that more explicitly than is currently conveyed.

While the manuscript is generally well-written, it could benefit from additional contextual discussion at some places and some editorial enhancements as noted below:

- 1) The format of the references is mixed – it should be harmonized.
- 2) It would be useful to discuss in more detail why the authors embarked on the development of the numerical model and the theoretical calculations to replicate the laboratory experiments. While it certainly is a worthwhile academic undertaking, greater discussion on what aspects of the laboratory experiments were complemented by these and the additional insights gained through the theoretical calculations would be useful for the readers.
- 3) Line 212: the sentence “In this analysis” ends awkwardly and needs to be reworded.
- 4) Equation 11 is missing or the numbering starting with the current equation 12 needs adjusting.
- 5) Please make the x- and y-axis units on Figures 7-9 consistent with those in Figures 4-6.

Decision letter (RSOS-192075.R0)

18-Feb-2020

Dear Dr Levendis,

The Subject Editor assigned to your paper ("A Simple Experiment on Global Warming") has now received comments from reviewers. We would like you to revise your paper in accordance with the referee and Associate Editor suggestions which can be found below (not including confidential reports to the Editor). Please note this decision does not guarantee eventual acceptance.

Please submit a copy of your revised paper before 12-Mar-2020. Please note that the revision deadline will expire at 00.00am on this date. If we do not hear from you within this time then it will be assumed that the paper has been withdrawn. In exceptional circumstances, extensions may be possible if agreed with the Editorial Office in advance. We do not allow multiple rounds of revision so we urge you to make every effort to fully address all of the comments at this stage. If deemed necessary by the Editors, your manuscript will be sent back to one or more of the original reviewers for assessment. If the original reviewers are not available we may invite new reviewers.

To revise your manuscript, log into <http://mc.manuscriptcentral.com/rsos> and enter your Author Centre, where you will find your manuscript title listed under "Manuscripts with Decisions." Under "Actions," click on "Create a Revision." Your manuscript number has been

appended to denote a revision. Revise your manuscript and upload a new version through your Author Centre.

When submitting your revised manuscript, you must respond to the comments made by the referees and upload a file "Response to Referees" in "Section 6 - File Upload". Pay special attention to the reviewers' comments concerning the wider applicability of your results and the extent to which your findings are unique. Please use this to document how you have responded to each of the comments, and the adjustments you have made. In order to expedite the processing of the revised manuscript, please be as specific as possible in your response.

- Ethics statement

- Data accessibility

If you wish to submit your supporting data or code to Dryad (<http://datadryad.org/>), or modify your current submission to dryad, please use the following link:
<http://datadryad.org/submit?journalID=RSOS&manu=RSOS-192075>

- Competing interests

- Authors' contributions

- Acknowledgements

- Funding statement

Kind regards,

Andrew Dunn

on behalf of Prof Jon Blundy (Subject Editor)

Reviewer comments to Author:

Reviewer: 2

Comments to the Author(s)

The paper describes a simple experiment that demonstrates that a CO₂ atmosphere cools more slowly than an O₂/N₂ atmosphere.

This is a revision of a paper that I had previously reviewed. I criticized the previous version because it did not reproduce the actual physics of the Earth's greenhouse effect. To the authors' credit, this version is improved. In particular, they have improved the discussion of the physics of what's going on in our climate system and seem to recognize the shortcomings of their experiment.

In the end, however, I still don't think this experiment reproduces the actual physics of the greenhouse effect. If you go to line 328 of the paper, where it refers to tables 2 and 3, they show that the main difference between heat loss in air and CO₂ is in the amount convective heat transport. This is not how CO₂ warms the Earth, which is mainly by modifying radiation to space (they know this, see line 336).

So I agree that they have shown that CO₂ reduces the cooling rate in their experiment, but the applicability to the present climate problem seems to me to be limited.

One way to fix this would be to put a prominent disclaimer up front in the paper that says that the mechanism here is not the same as those that are driving climate change. I think that would remove any scientific problems I have with this paper. However, this would create another problem. If this experiment is not applicable to the Earth, then what's the point of publishing the paper? Perhaps the authors could figure out a way to finesse that ... ?

Reviewer: 3

Comments to the Author(s)

This manuscript describes laboratory and numerical experiments to demonstrate the impacts of presence of CO₂ in the Earth's atmosphere on the earth-atmosphere energy exchange. The authors placed a heating element in an inflatable balloon and examine the time it takes to cool the element back to ambient temperature in the presence and absence of CO₂ (akin to examining surface temperature in the earth's atmosphere). Not surprisingly, they demonstrate that the

radiative heat loss from the element decreased in the presence of CO₂. As such the experiment could be regarded as a simple demonstration of the effects of the presence of CO₂ on perturbing the earth-atmosphere radiation budget and the atmosphere's subsequent warming, but not necessarily a rigorous representation of the magnitude of the warming or the complex interactions resulting in climate change. A quick search of the internet yields links to experiments which provide similar qualitative inferences on the effects of CO₂ and its greenhouse effect, for example:

<https://www.rsc.org/Education/Teachers/Resources/jesei/co2green/home.htm>

http://www.carboeurope.org/education/CS_Materials/Bernd-BlumeExperiments.pdf

Perhaps it could be argued that experiments presented here are more rigorous and controlled than the two examples above and that the authors have also developed an energy exchange mathematical model to capture and corroborate their experimental set-up. Climate change is one of the most prominent and debated societal issues currently; significant scientific and political debate exists on the magnitude and causes for the observed atmospheric warming trends. Thus, simple representations of such complex phenomena are useful to illustrate the likely impacts of rising atmospheric CO₂ concentrations. While the experiments presented do demonstrate CO₂ induced warming and the presentation of the mathematical framework is a useful contribution, I struggled to clearly identify the unique scientific contribution of this work – perhaps the authors can state that more explicitly than is currently conveyed.

While the manuscript is generally well-written, it could benefit from additional contextual discussion at some places and some editorial enhancements as noted below:

- 1) The format of the references is mixed – it should be harmonized.
- 2) It would be useful to discuss in more detail why the authors embarked on the development of the numerical model and the theoretical calculations to replicate the laboratory experiments. While it certainly is a worthwhile academic undertaking, greater discussion on what aspects of the laboratory experiments were complemented by these and the additional insights gained through the theoretical calculations would be useful for the readers.
- 3) Line 212: the sentence “In this analysis” ends awkwardly and needs to be reworded.
- 4) Equation 11 is missing or the numbering starting with the current equation 12 needs adjusting.
- 5) Please make the x- and y-axis units on Figures 7-9 consistent with those in Figures 4-6.

Author's Response to Decision Letter for (RSOS-192075.R0)

See Appendix D.

RSOS-192075.R1 (Revision)

Review form: Reviewer 2

Is the manuscript scientifically sound in its present form?

Yes

Are the interpretations and conclusions justified by the results?

Yes

Is the language acceptable?

Yes

Do you have any ethical concerns with this paper?

Yes

Have you any concerns about statistical analyses in this paper?

No

Recommendation?

Accept with minor revision (please list in comments)

Comments to the Author(s)

Overall, I think the authors now provide sufficient caveats to their analysis that I am happy with the paper.

Review form: Reviewer 3

Is the manuscript scientifically sound in its present form?

Yes

Are the interpretations and conclusions justified by the results?

Yes

Is the language acceptable?

Yes

Do you have any ethical concerns with this paper?

No

Have you any concerns about statistical analyses in this paper?

No

Recommendation?

Accept with minor revision (please list in comments)

Comments to the Author(s)

The authors have satisfactorily addressed most of my comments on the previous version of this manuscript. The additions to the manuscript help better explain the simplicity of the experiment relative to the complex earth-atmosphere climate system. I have a few additional minor suggestions for the authors consideration:

- 1) L53: "at 12-20" should perhaps be "at wavelengths of 12-20"
- 2) L54: "average Earth's surface temperature" should perhaps be "Earth's average surface temperature"
- 3) L84-86: this discussion should perhaps be expanded to convey that even though the bench-top experiment precludes a precise analogy with the complex global warming phenomena,

the experiments do demonstrate the effects of the presence of CO₂ on outgoing radiation from the Earth's surface.

Decision letter (RSOS-192075.R1)

Dear Dr Levendis:

On behalf of the Editors, I am pleased to inform you that your Manuscript RSOS-192075.R1 entitled "A Simple Experiment on Global Warming" has been accepted for publication in Royal Society Open Science subject to minor revision in accordance with the referee suggestions. Please find the referees' comments at the end of this email.

The reviewers and Subject Editor have recommended publication, but also suggest some minor revisions to your manuscript. Therefore, I invite you to respond to the comments and revise your manuscript.

- Ethics statement

- Data accessibility

If you wish to submit your supporting data or code to Dryad (<http://datadryad.org/>), or modify your current submission to dryad, please use the following link:
<http://datadryad.org/submit?journalID=RSOS&manu=RSOS-192075.R1>

- Competing interests

- Authors' contributions

- Acknowledgements

- Funding statement

Because the schedule for publication is very tight, it is a condition of publication that you submit the revised version of your manuscript before 06-Aug-2020. Please note that the revision deadline will expire at 00.00am on this date. If you do not think you will be able to meet this date please let me know immediately.

Reviewer comments to Author:

Reviewer: 3

Comments to the Author(s)

The authors have satisfactorily addressed most of my comments on the previous version of this manuscript. The additions to the manuscript help better explain the simplicity of the experiment relative to the complex earth-atmosphere climate system. I have a few additional minor suggestions for the authors consideration:

- 1) L53: "at 12-20" should perhaps be "at wavelengths of 12-20"
- 2) L54: "average Earth's surface temperature" should perhaps be "Earth's average surface temperature"
- 3) L84-86: this discussion should perhaps be expanded to convey that even though the bench-top experiment precludes a precise analogy with the complex global warming phenomena, the experiments do demonstrate the effects of the presence of CO₂ on outgoing radiation from the Earth's surface.

Reviewer: 2

Comments to the Author(s)

Overall, I think the authors now provide sufficient caveats to their analysis that I am happy with the paper.

Author's Response to Decision Letter for (RSOS-192075.R1)

See Appendix E.

Decision letter (RSOS-192075.R2)

Dear Dr Levendis,

It is a pleasure to accept your manuscript entitled "A Simple Experiment on Global Warming" in its current form for publication in Royal Society Open Science.

Appendix A

Reviewer's Comments to Authors of "A Simple Experiment on Global Warming"

1. The sentence beginning on line 101 appears to be repeated on line 112.
2. **Line 128** contains a typographical error...adsorbing should be absorbing.
3. Suggest adding a comma between balloon and pressurization in **line 161** to enhance clarity.
4. In **line 178** should r be r_e ?
5. Define T in **Eq. 3**. It is clearly the instantaneous heater temperature. The subscripts on the mass and heat capacity of the heater do not carry through consistently to Eq. 11 nor to the values listed in Table 1.
6. In **Table 1**: L_c appears to be simply V/A so is clearly not the same as in Eq. 2, which is claimed in line 204.
7. **Eq 4**. contains the heating element temperature. This should be the same T as that in Eq. 3. Elsewhere it appears that T_{he} refers to the constant initial heater temperature not the time dependent temperature. Also, the two T 's in Eq. 4 are reversed...written as it is, the convective heat loss would be positive when the surrounding gas is hotter than the heater!
8. **Eq. 6**: Although undefined, α here clearly refers to the thermal diffusivity, but later (for example in line 237) it becomes the absorptivity. Also, the subscript on L appears to be uppercase while in line 204, Table 1 and Eq. 5 it is lower case.
9. The entire section on **Radiative Energy Losses** needs editing. The background for **Eq 7** could be expanded to give some sense of how the physical processes of gaseous absorption and emission are being modelled for the work to be more accessible to the intended audience as articulated in the Abstract and Introduction. It would be useful to include how the energy balance on the heater element is carried out while incorporating the approximate treatment of gaseous radiation.
 - a. **Eq. 9**: Clarify the meanings of the two temperatures in the ratio; it is unclear what the subscript "air" refers to since air is one of the two gases tested. Also in the treatment of gas radiation used here as developed by Hottel the ratio is raised to a (non-unity) power depending on the gas. The absence of a power here may be a typo.
 - b. **Eq. 10**: This "equivalent length" is presumably the "mean beam length" of the Hottel treatment. It is unclear how the numerical prefix (0.167) was selected.
 - c. It is unclear how h_{rad} for CO_2 can be an order of magnitude lower than h_{rad} for air. The only difference seems to be the value of the gas absorptivity which is zero for air and nearly zero (0.096) for carbon dioxide as shown by the values in Table 3.

- d. **Table 3** values need to be checked. In particular the values in line 3 appears not to have the units listed since the numerical values are clearly in units of *atm-m* if line 2 is correctly in *m* not *ft*.
 - e. **Equation 11** has some spurious subscripts, probably typographical. For example ε_b should be ε_{he} and h should be h_{conv} for consistency with Eqs. 3, 4, and 7. It also appears that T_i should be T_∞^4
10. **Eq 12.** Again the subscript on k should be changed from b to he to be consistent with the nomenclature used in the display equations, although Table 1 lists the thermal and radiative properties of the heater with no subscripts at all.

Appendix B

Dear Dr. Kennett,

We received two excellent reviews of our manuscript entitled “A Simple experiment on Global Warming” and we revised it accordingly. Please find herein the revised manuscript and the rebuttal.

Thank you for your attention,

Cordially,

Yiannis A. Leventis, PhD, FRSC

College of Engineering Distinguished Professor
Director of the Combustion and Air Pollution Laboratory
Fellow of the ASME, SAE, RSC and of the Combustion Institute
Department of Mechanical and Industrial Engineering
334 SN, Northeastern University
360 Huntington Ave., Boston, MA 02115

Reviewer: 1

This review is impressively thorough. We want to thank the reviewer for checking every equation and the calculations, and alerting us on all deficiencies.

These experiments appear to be accessible to an educated non-expert audience in line with the Introduction. The effect of a CO₂ environment on reducing the cooling rate of a heated immersed object below that measured in air is clear and well discussed. The analytical section is insufficiently detailed and too disjointed to achieve the same end.

We agree with the reviewer. We went over the analytical section of the manuscript and thoroughly revised it for clarity and detail.

The Hottel method for dealing with gaseous radiation is approximate but easy to understand, however the narrative does not attempt to describe the Hottel method, even in a cursory fashion, nor to make clear how the energy balance was done and how the Hottel absorptivity, emissivity and mean beam length (or average mean beam length) fit in.

We included a write-up summarizing Hottel’s method in the revised manuscript, see lines 263-277.

My attached comments point out several typographical errors as well as inconsistencies in notation.

Reviewer's Comments to Authors of "A Simple Experiment on Global Warming"

1. The sentence beginning on line 101 appears to be repeated on line 112.

We agree with the reviewer, and we now have removed the duplicate phrase.

2. Line 128 contains a typographical error...adsorbing should be absorbing.

We appreciate that the reviewer noticed this. The correction was made.

3. Suggest adding a comma between balloon and pressurization in line 161 to enhance clarity. This correction was made.

4. In line 178 should r be re ? This correction was made.

5. Define T in Eq. 3. It is clearly the instantaneous heater temperature. The subscripts on the mass and heat capacity of the heater do not carry through consistently to Eq. 11 nor to the values listed in Table 1. This correction was made as T_{heater} .

6. In Table 1: L_c appears to be simply VA so is clearly not the same as in Eq. 2, which is claimed in line 204. The Characteristic Length in Table 1 and in Eq. 5 of the revised manuscript are now consistent and L_c has been replaced with $2r_e$, which is the diameter of an equivalent sphere for the heater.

7. Eq 4. contains the heating element temperature. This should be the same T as that in Eq. 3. Elsewhere it appears that T_{he} refers to the constant initial heater temperature not the time dependent temperature. Also, the two T s in Eq. 4 are reversed...written as it is, the convective heat loss would be positive when the surrounding gas is hotter than the heater!

We agree with the reviewer. The questioned equation has been now correctly given as:

$\dot{Q}_{\text{convection}} = h_{\text{conv}}A_{\text{heater}}(T_{\text{heater}} - T_{\text{gas}})$, which is Eq. 3 in the revised manuscript.

8. Eq. 6: Although undefined, α here clearly refers to the thermal diffusivity, but later (for example in line 237) it becomes the absorptivity. Also, the subscript on L appears to be uppercase while in line 204, Table 1 and Eq. 5 it is lower case.

We agree with the reviewer. The thermal diffusivity is now defined with α_d and the gas absorptivity is defined by α_{gas} .

9. The entire section on Radiative Energy Losses needs editing. The background for Eq 7 could be expanded to give some sense of how the physical processes of gaseous absorption and emission are being modelled for the work to be more accessible to the intended audience as articulated in the Abstract and Introduction. It would be useful to include how the energy balance on the heater element is carried out while incorporating the approximate treatment of gaseous radiation.

We agree with the reviewer. We carefully edited this section in its entirety for clarity.

- a. **Eq. 9:** Clarify the meanings of the two temperatures in the ratio; it is unclear what the subscript “air” refers to since air is one of the two gases tested. Also in the treatment of gas radiation used here as developed by Hottel the ratio is raised to a (non-unity) power depending on the gas. The absence of a power here may be a typo.
We agree with the reviewer. The absence of the power was a typo. This has now been corrected. Also the denominator of the fraction in (what is now Eq. 8) in the revised manuscript is T_{heater} .
- b. **Eq. 10:** This “equivalent length” is presumably the “mean beam length” of the Hottel treatment. It is unclear how the numerical prefix (0.167) was selected.
The equivalent length is indeed the mean beam length. Hence, for clarity, the questioned parameter L_e has now been replaced with L_{m-b} . There was a typo in Eq. 10 of the submitted manuscript. The corrected equation is now Eq. 9 in the revised manuscript as follows:
$$L_{m-b} = 3.66 \frac{V_{chamber}}{A_{chamber}} = 3.66(0.167)d = 0.61 d$$
- c. It is unclear how h_{rad} for CO_2 can be an order of magnitude lower than h_{rad} for air. The only difference seems to be the value of the gas absorptivity which is zero for air and nearly zero (0.096) for carbon dioxide as shown by the values in Table 3.
We agree, we went over all values in Table 3 and use of consistent calculations based on the multiple technical reports on this project yielded values of the radiative heat transfer coefficients of $h_{rad} = 5.74 \text{ W/m}^2\text{K}$ for air and $h_{rad} = 4.9\text{-}4.78 \text{ W/m}^2\text{K}$ for CO_2 , depending on the size of the balloon. These values are consistent with the reviewer’s comment. The radiative heat transfer coefficient was not used in the simulation model, Eq. 10.
- d. Table 3 values need to be checked. In particular the values in line 3 appears not to have the units listed since the numerical values are clearly in units of atm-m if line 2 is correctly in m not ft.
We agree with the reviewer. The units in the revised manuscript are now consistent with the tables reported by Hottel.
- e. Equation 11 has some spurious subscripts, probably typographical. For example ε_b should be ε_{he} and h should be h_{conv} for consistency with Eqs. 3, 4, and 7. It also appears that T_i should be T_{∞}
We agree with the reviewer. The issues in Eq. 11 of the submitted manuscript have now been corrected in Eq. 10 of the revised manuscript.
10. **Eq 12.** Again the subscript on k should be changed from b to he to be consistent with the nomenclature used in the display equations, although Table 1 lists the thermal and radiative properties of the heater with no subscripts at all. We agree with the reviewer, the change of the subscript has been made.

Reviewer: 2

We want to thank the reviewer for being critical of our experimental approach. It made us re-examine our approach and explain more thoroughly why we believe that such an experiment is valid.

This paper is a description of experiment designed to replicate the greenhouse effect and demonstrate the warming power of carbon dioxide. I look favorably upon their goal, but unfortunately their experiment does not reproduce the physics of the greenhouse effect.

The main problem with this paper is that their experiment does simulate the physics of the greenhouse effect. The greenhouse effect works because there is a temperature gradient in the atmosphere — the upper atmosphere is colder than the lower atmosphere.

We appreciate the favorable attitude of the reviewer regarding our goal, and we welcome the criticism. We also welcome the references that the reviewer provided us. In defense of our approach we wrote the following explanation inserted at the end of the Introduction section of the revised manuscript:

In this experiment, Earth is approximated with the small heater enclosed in a much larger gas-filled spherical balloon, which approximates the Earth's atmosphere. The balloon is filled with either air or carbon dioxide. The heater is deliberately brought to Earth-like temperatures. The heater's surrounding gas also heats up and a negative temperature gradient develops in the gas inside the balloon, in the outward radial direction. This temperature gradient in the gas is akin to the negative temperature gradient with increasing altitude inside Earth's Troposphere. Upon reaching a predetermined temperature (50 °C), the electricity is turned off and the heater is allowed to cool by losing energy (heat) to the balloon environment. Heat transfer takes place by convection to the gas inside the balloon and by radiation to its surroundings in the far-infrared wavelength spectrum. This is similar to Earth's case. A portion of the long wavelength radiation is absorbed by the gas inside the balloon when it contains infrared radiation (IR) absorbing molecules, such as CO₂. Another portion of the long wave radiation is transmitted through the gas and then it is mostly transmitted outwards through the membrane of the balloon. A fraction is likely absorbed by the balloon. This is analogous to what occurs in the Earth's atmosphere, where radiation emitted by the surface of the Earth is absorbed by infrared radiation-absorbing molecules in the atmosphere, clouds or other aerosols [22]. A fraction of the emitted radiation from the Earth's surface escapes to outer space through "the atmospheric window".

The Earth's atmosphere emits radiation into a spherical shell. A portion of this emitted radiation is directed back to Earth and a portion is directed to outer space. A similar heat transfer process occurs in the heater/balloon experiment. Although this gas emission differs from the radiation emitted by a black body; it is often found convenient in radiation calculations to treat gas emissions as a blackbody at some lower effective temperature. This temperature is called the effective sky temperature [23], and it is the temperature of the I-R-absorbing species in the atmosphere. The radiation emitted from these species to space will be lower than that from the

Earth's surface to space, had those I-R species been absent. This is because the temperature lapse rate (temperature gradient) in the atmosphere is negative ($-6.5\text{ }^{\circ}\text{C}/\text{km}$), hence, the IR-absorbing species are at a lower temperature than the Earth's surface. This produces a planetary energy imbalance, which leads to warming [24].

If you add carbon dioxide, you raise the effective radiating level of the atmosphere, so that the atmosphere is then radiating from a colder temperature. This produces a planetary energy imbalance, which leads to warming. This is such basic stuff that I couldn't find a good peer-reviewed paper to cite. So I would point them to these two links:

https://nam05.safelinks.protection.outlook.com/?url=https%3A%2F%2Fskepticalscience.com%2Fbasics_one.html&data=02%7C01%7CY.Levendis%40northeastern.edu%7C61b9eacf85784043787808d711a47057%7Ca8eec281aaa34daeac9b9a398b9215e7%7C0%7C0%7C636997270294102794&data=6N2uLPJGA8KLUHkTbj7vtDPuUqIPbXBLTA%2B%2B6%2F4DZA%3D&reserved=0

or

<https://nam05.safelinks.protection.outlook.com/?url=https%3A%2F%2Fwww.youtube.com%2Fwatch%3Fv%3D4PAbm1u1IVg&data=02%7C01%7CY.Levendis%40northeastern.edu%7C61b9eacf85784043787808d711a47057%7Ca8eec281aaa34daeac9b9a398b9215e7%7C0%7C0%7C636997270294102794&data=geHMZ5pfQfOicB6FUSkmjZ6GeHa93pcYIS82hvhLW1I%3D&reserved=0>

This is also qualitatively the case in our experiment, as mentioned above.

As a result, while their experiment gets the right answer (carbon dioxide warms the climate), it does so for the wrong reason. Thus, I don't think this paper is publishable in its present form. Our experiment is not contradicting the notion that the accumulating CO in the atmosphere "raises the effective radiating level of the atmosphere, so that the atmosphere is then radiating from a colder temperature". Both Earth's atmosphere and the atmosphere in the balloon experience decreasing temperatures in the radial direction, i.e., temperatures decreasing with altitude.

A few more minor comments:

Line 54: "However, the net influence of the water vapor on atmospheric temperature is unclear." No, water vapor is responsible for the majority of the greenhouse effect. See this paper:

<https://nam05.safelinks.protection.outlook.com/?url=https%3A%2F%2Fagupubs.onlinelibrary.wiley.com%2Fdoi%2F10.1029%2F2010JD014287&data=02%7C01%7CY.Levendis%40northeastern.edu%7C61b9eacf85784043787808d711a47057%7Ca8eec281aaa34daeac9b9a398b9215e7%7C0%7C0%7C636997270294112786&data=pmhEia5eLaZPU0VN9Tw4kcaOJGnjPlvTkgb1Jf7qG2Y%3D&reserved=0>

Thank you for this information. The phrase: "However, the net influence of the water vapor on atmospheric temperature is unclear." has been removed, and the following paragraph has been added on line 56 of the Introduction Section of the revised manuscript:

"Water vapor absorbs terrestrial radiation like the other greenhouse gases and produces a warming effect. However, there is also a cooling effect. As the water vapor rises in the atmosphere, it condenses and forms clouds. Clouds reflect solar radiation and reduce the heating of the earth. It has been reported that water vapor is the dominant contributor to the global

greenhouse effect (~50% of the effect), followed by clouds (~25%) and then CO₂ with ~20%. Other absorbers play minor roles [7]. Nevertheless, as the water cycle of the atmosphere is a naturally-occurring phenomenon, this work is not focusing on its effects on global warming.”

I found the nomenclature of T(infinity) being the initial temperature to be confusing. It seems like that should be the long-time equilibrium temperature.

T_{infinity} has now been replaced by T_{gas}

Eq. 11 jumped out of nowhere. I'm not sure how you got that from the previous equations. This has now been remedied by adjusting the subscripts of all equations and correcting typing errors. It should be clear now. We regret the lack of clarity in the submitted manuscript.

I feel bad being so negative and I encourage the authors to keep working on producing a good simulation of the greenhouse effect.

Your criticism is welcome as it helped us strengthen this manuscript.

Appendix C

Dear Dr. Kennett,

We received two excellent reviews of our manuscript entitled “A Simple experiment on Global Warming” and we revised it accordingly. Please find herein the revised manuscript and the rebuttal.

Thank you for your attention,

Cordially,

Yiannis A. Leventis, PhD, FRSC

College of Engineering Distinguished Professor
Director of the Combustion and Air Pollution Laboratory
Fellow of the ASME, SAE, RSC and of the Combustion Institute
Department of Mechanical and Industrial Engineering
334 SN, Northeastern University
360 Huntington Ave., Boston, MA 02115

Appendix D

Dear Prof. Blundy and Mr. Dunn,

We received two constructive reviews of our manuscript entitled “A Simple experiment on Global Warming” and we revised it accordingly. Please find herein the revised manuscript and the rebuttal.

Thank you for your attention,

Cordially,

Yiannis A. Leventis, PhD, FRSC

College of Engineering Distinguished Professor
Director of the Combustion and Air Pollution Laboratory
Fellow of the ASME, SAE, RSC and of the Combustion Institute
Department of Mechanical and Industrial Engineering
334 SN, Northeastern University
360 Huntington Ave., Boston, MA 02115

Reviewer: 2

Comments to the Author(s)

The paper describes a simple experiment that demonstrates that a CO₂ atmosphere cools more slowly than an O₂/N₂ atmosphere.

This is a revision of a paper that I had previously reviewed. I criticized the previous version because it did not reproduce the actual physics of the Earth’s greenhouse effect. To the authors' credit, this version is improved. In particular, they have improved the discussion of the physics of what’s going on in our climate system and seem to recognize the shortcomings of their experiment.

In the end, however, I still don't think this experiment reproduces the actual physics of the greenhouse effect. If you go to line 328 of the paper, where it refers to tables 2 and 3, they show that the main difference between heat loss in air and CO₂ is in the amount convective heat transport. This is not how CO₂ warms the Earth, which is mainly by modifying radiation to space (they know this, see line 336).

We have inserted lines 347-354 to address the convolution of convection and gaseous radiation in this work, as follows: “This relatively large convective loss contribution is due to the scale limits of a benchtop experiment in not duplicating the dominance of radiation on climate change. Even so, the simulation predicts that the radiative heat loss decreased by 17% in the carbon dioxide filled balloon, as compared to that in the air filled balloon. The large fraction of total energy loss by convection in the experiment is more consistent with the combined heat transfer

mechanisms in the troposphere that contribute to the temperature profile therein. The global energy exchange between the Earth and space is dominated by the radiative heat transfer.”

So I agree that they have shown that CO₂ reduces the cooling rate in their experiment, but the applicability to the present climate problem seems to me to be limited.

One way to fix this would be to put a prominent disclaimer up front in the paper that says that the mechanism here is not the same as those that are driving climate change. I think that would remove any scientific problems I have with this paper. However, this would create another problem. If this experiment is not applicable to the Earth, then what's the point of publishing the paper? Perhaps the authors could figure out a way to finesse that ... ?

We have inserted a disclaimer that our experiment is not an exact analogy with climate change, see lines 83-86 of the revised manuscript. However, the contribution of the paper is the use of cooling rates in the experiment that demonstrates the effects of carbon dioxide in an atmosphere in combination with an analytical simulation that is consistent with the experimental results.

Reviewer: 3

Comments to the Author(s)

This manuscript describes laboratory and numerical experiments to demonstrate the impacts of presence of CO₂ in the Earth's atmosphere on the earth-atmosphere energy exchange. The authors placed a heating element in an inflatable balloon and examine the time it takes to cool the element back to ambient temperature in the presence and absence of CO₂ (akin to examining surface temperature in the earth's atmosphere). Not surprisingly, they demonstrate that the radiative heat loss from the element decreased in the presence of CO₂. As such the experiment could be regarded as a simple demonstration of the effects of the presence of CO₂ on perturbing the earth-atmosphere radiation budget and the atmosphere's subsequent warming, but not necessarily a rigorous representation of the magnitude of the warming or the complex interactions resulting in climate change. A quick search of the internet yields links to experiments which provide similar qualitative inferences on the effects of CO₂ and its greenhouse effect, for example:

<https://nam05.safelinks.protection.outlook.com/?url=https%3A%2F%2Fwww.rsc.org%2FEducation%2FTeachers%2FResources%2Fjesei%2Fco2green%2Fhome.htm&data=02%7C01%7CY.Levendis%40northeastern.edu%7Ce706d947f7a84278f98308d7b45e9cf6%7Ca8eec281aaa34daeac9b9a398b9215e7%7C0%7C0%7C637176191594641666&sdata=Uu8QoWzfPbfb7K7h3li7oOCu16A7neQNYn4wBd3kH4w%3D&reserved=0>

https://nam05.safelinks.protection.outlook.com/?url=http%3A%2F%2Fwww.carboeurope.org%2FEducation%2FCS_Materials%2FBernd-BlumeExperiments.pdf&data=02%7C01%7CY.Levendis%40northeastern.edu%7Ce706d947f7a84278f98308d7b45e9cf6%7Ca8eec281aaa34daeac9b9a398b9215e7%7C0%7C0%7C637176191594641666&sdata=LixV5xLCGCXaj38%2FgFgLCZlwrAjwz39rR1FEGDAfqGw%3D&reserved=0

Perhaps it could be argued that experiments presented here are more rigorous and controlled than the two examples above and that the authors have also developed an energy exchange mathematical model to capture and corroborate their experimental set-up.

We agree with the assessments of this referee. We also referenced the aforementioned links. The inclusion of a rigorous analytical model of the cooling rates separates the present work from these references.

Climate change is one of the most prominent and debated societal issues currently; significant scientific and political debate exists on the magnitude and causes for the observed atmospheric warming trends. Thus, simple representations of such complex phenomena are useful to illustrate the likely impacts of rising atmospheric CO₂ concentrations. While the experiments presented do demonstrate CO₂ induced warming and the presentation of the mathematical framework is a useful contribution, I struggled to clearly identify the unique scientific contribution of this work – perhaps the authors can state that more explicitly than is currently conveyed.

The scientific contribution of this work is the combination of an experiment that demonstrates the effect of carbon dioxide on an atmosphere and an analytical simulation that is consistent with the experimental results. The paper includes the physics of a participating medium on radiative heat transfer that is a foundation of climate models. This is now mentioned in Line 357.

While the manuscript is generally well-written, it could benefit from additional contextual discussion at some places and some editorial enhancements as noted below:

1) The format of the references is mixed – it should be harmonized.

We agree. We have remedied this issue in this revision of the manuscript.

2) It would be useful to discuss in more detail why the authors embarked on the development of the numerical model and the theoretical calculations to replicate the laboratory experiments. While it certainly is a worthwhile academic undertaking, greater discussion on what aspects of the laboratory experiments were complemented by these and the additional insights gained through the theoretical calculations would be useful for the readers.

We agree. The following phrases were added to the Introduction Section of the revised manuscript:

Lines 70-74: To fill this gap, a basic benchtop experiment was designed and complemented with a theoretical analysis.

Lines 80-83: The scale of the benchtop experiment precludes an exact analogy with global warming, because of the combined radiative and convective heat transfer mechanism. The

theoretical analysis confirms the cooling behavior and the expected magnitude of different cooling rates with and without carbon dioxide present.

3) Line 212: the sentence "In this analysis ..." ends awkwardly and needs to be reworded. We agree. This sentence was fixed as follows in lines 219-221: In this analysis, the lumped capacitance method outlined in Ref. [29] was used because the spatial temperature gradients within the heater were considered to be negligible.

4) Equation 11 is missing or the numbering starting with the current equation 12 needs adjusting.

We agree. This equation numbering problem has now been fixed.

5) Please make the x- and y-axis units on Figures 7-9 consistent with those in Figures 4-6. This review is impressively thorough. We want to thank the reviewer for checking every equation and the calculations, and alerting us on all deficiencies.

We agree. This has now need fixed. The units in all these six plots have been harmonized.

Appendix E

Dear Prof. Blundy and Mr. Dunn,

We received two final reviews of our manuscript entitled “*A Simple experiment on Global Warming*” and we revised it accordingly. Please find herein the newly revised manuscript and the rebuttal.

Thank you for your attention,

Cordially,

Yiannis A. Levendis, PhD, FRSC

College of Engineering Distinguished Professor
Director of the Combustion and Air Pollution Laboratory
Fellow of the ASME, SAE, NAI, RSC and of the Combustion Institute
Department of Mechanical and Industrial Engineering
334 SN, Northeastern University
360 Huntington Ave., Boston, MA 02115

Reviewer: 2

Overall, I think the authors now provide sufficient caveats to their analysis that I am happy with the paper.

We want to thank this reviewer for his past comments and for looking over our manuscript one more time.

Reviewer: 3

The authors have satisfactorily addressed most of my comments on the previous version of this manuscript. The additions to the manuscript help better explain the simplicity of the experiment relative to the complex earth-atmosphere climate system. I have a few additional minor suggestions for the authors consideration:

- 1) L53: “at 12-20” should perhaps be “at wavelengths of 12-20”
Done.
- 2) L54: “average Earth’s surface temperature” should perhaps be “Earth’s average surface temperature”
Done.
- 3) L84-86: this discussion should perhaps be expanded to convey that even though the bench-top experiment precludes a precise analogy with the complex global warming phenomena, the experiments do demonstrate the effects of the presence of CO₂ on outgoing radiation from the Earth’s surface. We agree. As recommended by the reviewer this text has now been modified to read:
“The scale of the benchtop experiment precludes an exact analogy with global warming, because of the combined radiative and convective heat transfer mechanism, **the experiments do demonstrate the effects of the presence of CO₂ on outgoing radiation from the Earth’s surface. Moreover,** the theoretical analysis confirms the cooling behavior and the expected magnitude of different cooling rates with and without carbon dioxide.” This text appears in lines 88-88 of the newly revised manuscript, the highlighted text was added.

We want to thank this reviewer for his past and present comments and for looking over our manuscript one more time.